# A New Protocol to Mitigate Damage to Germination Caused by Black Layers in Maize (*Zea mays* L.) Seeds

Joon Ki Hong [ID], Jeongho Baek [ID], Sang Ryeol Park [ID], Gang Seob Lee [ID] and Eun Jung Suh *

Gene Engineering Division, National Institute of Agricultural Sciences, Rural Development Administration, 370 Nongsaengmyeong-ro, Jeonju 54874, Republic of Korea; jkhongok@korea.kr (J.K.H.); firstleon@korea.kr (J.B.); srpark@korea.kr (S.R.P.); kangslee@korea.kr (G.S.L.)
* Correspondence: seji00@korea.kr; Tel.: +82-63-238-4660

**Abstract:** Maize seeds harvested in the field have higher vitality than those harvested in greenhouses but have higher contamination rates in terms of fungal or bacterial infection. It is important to disinfect maize seeds before sowing because seeds are a source of infection and damage crop production. In this study, we aimed to provide an efficient seed sterilization method to manage fungal or bacterial infections of field-harvested maize seeds. The optimized sterilization protocol was set up according to the disinfectant liquid immersion time, inverting RPM (rotations per minute), number of seeds, and black layer removal. We put 20 grains of maize seeds in 100% commercial bleach disinfectant containing 4–5% NaClO and performed 20 min of inversion at 45 RPM. After standing without inverting for the next 25 min in the sterile hood, inversion at 45 RPM for another 40 min was performed. By using this protocol, microorganisms occurred at a low rate with an average of 11.7%. Moreover, it was shown that microorganisms occurred at the lowest rate (average of 0.29% of seeds) when the black layer was removed. In addition, this sterilization method did not affect the growth and development of maize plants. These results revealed that black layer removal from maize seeds is a highly efficient, easy, and inexpensive sterilization method and can be used for seeds of various maize lines.

**Keywords:** contamination; germination; field-harvested; maize seed; seed sterilization

## 1. Introduction

Seeds are the starting point for plant growth, and seed-mediated microorganisms are an important source of inoculation for plant-related biological communities [1]. Seeds are colonized by many bacteria and fungi, and some may be harmful for the germination and subsequent plant growth of seedlings [2,3]. However, it is never easy to effectively sterilize seeds without affecting the growth of plants [4,5]. Various factors, such as the type or origin of plants and the degree of contamination of bacteria and fungi, can affect seed sterilization; thus, a single disinfection method is not suitable for all plants [6]. Therefore, it is important to choose the type of disinfectant and sterilization conditions to minimize contamination and the negative effects caused by bacteria and fungi in seeds in order to induce high germination rates and appropriate plant growth [7,8].

Achieving high-quality seed sterilization is affected by various factors such as the type of disinfectant and the duration of sterilization [9]. Sodium hypochlorite (NaClO), ethanol, hydrogen peroxide ($H_2O_2$), and mercury chloride ($HgCl_2$) are substances commonly used for sterilization of the seeds of various crops [10–12]. Sodium hypochlorite (NaClO), with strong oxidative properties, is very effective in removing bacteria or reducing bacterial populations [12]. Hypochlorous acid (HOCl) formed by diluted hypochlorite can be converted into highly toxic chloramine (highly oxidative and diffuse) and enter through the cell membrane to react with components inside the cell [13]. In addition, it has been reported that the use of surfactant tween 20 with NaClO can increase the efficiency of

seed sterilization and improve wettability [7,14]. As a commonly used strong medical disinfectant, 70–75% ethanol penetrates the cell membrane and denatures various proteins, inactivating some bacteria; however, it is a strong phytotoxic agent [12]. When used for only a few minutes or even a few seconds before the use of any other disinfectant, the sterilization effect is higher than when only ethanol is used [12]. Hydrogen peroxide ($H_2O_2$) is a dual reactant involved in signal transmission and damage in plant development and physiological characteristics, and low concentrations of $H_2O_2$ have demonstrated sterilization activity without affecting seed growth and germination [15]. However, it was reported that the damage effect of $H_2O_2$ on tissue and growth may vary depending on the species of plant after sterilization, so appropriate concentrations should be used [16]. Mercuric chloride ($HgCl_2$) is a highly toxic chemical that is difficult to dispose of, and it therefore has not been widely used in seed sterilization recently [17]. However, most newly developed disinfectants cannot be used for many pathogens [6].

In the case of maize, field-harvested material has higher vitality than maize grown in the greenhouse, but with higher levels of contamination [6,7,18]. In addition, effective sterilization methods for seeds harvested in fields carrying soil-borne pathogens are still being studied [6,19]. Recently, maize seeds were sterilized with a 0.024–20% NaClO concentration for 20 min or sterilized with 0.1% and 10% $H_2O_2$ for 30 min [20–27]. $HgCl_2$ has been reported to sterilize maize seeds after treatment in a 1% solution for 2 min [28]. After these treatments, most seeds germinated within 10 days and did not damage plant growth. When sterilization time was increased by 0.5–5 h under shaking in a maize seed sterilization mixture using NaClO, the number of infected seeds decreased but the germination rate was low [29]. However, it is reported that the number of infected seeds decreased as a result of sterilization involving shaking for 2.5 h and 1 min of vacuum treatment [29]. The shaking added to the sterilization process ensured that the sterilization solution and maize seeds were mixed well and that the sterilization solution was sufficiently wet on the seed surface, showing a high sterilization effect by removing the bubbles of the seeds [29]. Among the various methods, an ethanol–NaClO-based treatment was reported to be suitable for maize seed sterilization according to efficiency, quantitative, and qualitative evaluation of the sterilization treatment using a multi-scale approach to determine reductions in microorganisms such as bacteria and fungi on maize seeds [19]. Despite careful attention during the application of sterilization techniques, they can result in 100% loss in field-grown plants due to bacteria and fungi that remain unremoved from the seeds [7,19,30].

When harvesting maize seeds, the black layer is a layer used to determine the maturity of maize because it means the maize has reached physiological maturity. However, it is easily exposed to bacterial and fungal contaminants due to the effects of rainy seasons and short-term torrential rains during the seed maturity stage. Recently, Korea's climate has also changed to a more humid subtropical climate due to irregular short-term torrential rains in the maize seed maturity stage (as can be seen on Weather Spark, https://weatherspark.com/countries/KR) [31]. As a result, the harvested maize seeds can be contaminated by various pathogens during the cultivation period. Therefore, it is important to sterilize the harvested seeds from field-grown plants by selecting an appropriate sterilization method [19,32]. Here, we report an efficient and reproducible sterilization method developed specifically by removing black layers from maize seeds grown and harvested in fields. This method was intended to increase the effect of sterilization by improving the seed sterilization of sodium hydrochlorite (NaClO), which is widely used as both a general disinfectant and endothelial treatment in some seeds. The seeds used in this study are widely used for research purposes, and the seed infection rate of pathogens was high when cultivated in the field. Therefore, this study was conducted to find the best and most efficient sterilization procedure which can be used in maize seeds, which was achieved by comparing various sterilization protocols used in different seeds.

## 2. Materials and Methods

### 2.1. Plant Materials

In this study, the KS140 and KS141 varieties were obtained from the NICS (National Institute of Crop Science, Rural Development Administration, 181 Hyeoksin-ro, Iseo-myeon, Wanju-Gun, Jeollabuk-do, Republic of Korea), the HW3 variety was procured from the MES (Maize Experiment Station, Gangwondo Agricultural Research and Extension Services, Republic of Korea), the A188, B73, B98, B104, H99, Hi IIA, and Hi IIA × Hi IIB varieties were acquired from the USDA (US Department of Agriculture) [33], and two F1 hybrid varieties (Hi IIA(♂) × B73(♀) and B73(♂) × Hi IIA(♀)) were produced through Hi IIA and B73 hybridization [32], with the 12 total varieties used for seed sterilization.

Twenty grains of seeds were placed on a 10 cm diameter Petri dish with filter paper and germinated under long-day conditions (16 h light/8 h dark) at 25 °C. To germinate maize seeds, 10 mL of distilled water was soaked in filter paper for germination, and distilled water was replenished daily to maintain humidity during germination. Two days after the transfer, it was transferred to a 50-hole pot with a bed soil and perlite mix at the ratio of 9:1 and then transplanted into the field when leaf length reached 15–20 cm. In addition, germination in the 50-hole pot was based on the appearance of seedlings of more than 5 mm, and germination was investigated from the second day to the seventh after transfer to the soil. Pollination was carried out at 18–22 °C between 8:00 and 10:00 in the morning, which is a timeframe known to have an optimal pollination rate in field conditions (Weather Spark, https://weatherspark.com/y/142288/Average-Weather-in-Jeonju-South-Korea-Year-Round, accessed on 2 January 2021) between June and July in Jeonju, Republic of Korea [33]. Although there are slight differences depending on the variety and environment, mature seeds were generally harvested from July to August (Figure S1).

### 2.2. Sterilization Procedure

To analyze sterilization efficiency when the sterilization time of seeds increases, twenty grains of seeds were placed in a 50 mL conical tube containing 35 mL of 100% commercial bleach (4–5% NaClO content) including 0.1% tween 20 before being shaken for 0 min (Treatment 1), 15 min (Treatment 2), and 30 min (Treatment 3), respectively, and then washed five times with sterile distilled water (Figure S2). Alternatively, the method of sterilizing at the same time as inverting in maize seed sterilization was modified, inverting (Vertical Rotator AG, FINEPCR, Gunpo, Republic of Korea) was performed at 45 RPM for 20 min, and the tube was then allowed to stand for 25 min (resting) in the sterile hood. Immediately afterwards, inverting was performed again at 45 RPM for 40 min (Treatment 4), and samples were washed five times with sterile distilled water for 2 min (Figure S2). The sterilized seeds were incubated under long-day conditions (16 h light/8 h dark) at 25 °C after placing them at 30~45 angles (Figure S3), thus allowing the black layer to touch the basal medium (MS basal salts including vitamins, 3% sucrose, and 0.3% Gelrite; 100 × 40 mm Petri dish). All experiments were repeated three times using more than 150 seeds. Germination was based on the appearance of roots or seedlings of more than 3 mm in seeds, and seed contamination was determined via bacteria or fungi around seeds that could be visually identified seven days after placing into the media investigated (Figure S4). Calculation of seed germination (%) and contamination (%) was performed according to previously described methods [19,31].

$$Seed germination\ (\%) = \left( \frac{Number\ of\ grown\ seeds}{Tatal\ number\ of\ seeds} \right) \times 100$$

$$Seed contamination\ (\%) = \left( \frac{Number\ of\ seeds\ with\ microbial\ growth}{Tatal\ number\ of\ seeds} \right) \times 100$$

### 2.3. Sterilization Efficiency of Black Layer Removal, Inverting RPM Change, and Number of Seeds

To investigate the effect of the black layer of maize seeds on sterilization, after removing 1~4 mm$^2$ of the black layer of the seed using sharp forceps (Figure S5), seeds were sterilized using the previous methods and the removed portion was moved to the MS medium and grown for 7 days to measure germination and contamination. Next, in order to analyze the effect of inverting RPM and the number of seeds in seed sterilization with the black layer removed using the method of Treatment 4, the inverting RPM was changed to 25, 45, and 65 to investigate germination and contamination, respectively. In addition, the number of maize seeds with the black layer removed was changed to 10, 20, and 30 grains, and sterilization was performed using the method of Treatment 4. Seed germination and contamination were performed according to methods previously described above.

### 2.4. Statistical Analysis

One-way ANOVA (Shapiro–Wilk normality statistic test) ($p < 0.01$) was performed for the measurement results of these experiments [32]. The Jamovi-2.3.26 (https://www.jamovi.org, accessed on 2 May 2023) program and SigmaPlot 12.5 (Systat Software, GmbH, Erkrath, Germany) were used for all statistical analyses [32]. This limited the range of values obtained from mean $\pm$ standard deviation (SD) or standard error (SE) with 95% confidence. In addition, the significance of differences in amounts between the control group and experimental group were tested using the Student's *t*-test ($p < 0.01$).

## 3. Results and Discussion

### 3.1. Selecting a Surface Sterilization Procedure

Ecology, physiology, and molecular biology studies are being actively conducted to develop maize varieties resistant to various environmental stresses [9]. This study was conducted to establish the optimal seed sterilization method for maize seeds using commercial disinfectants.

In this study, the effect of 100% commercial bleach (4–5% NaClO) on contamination and seed germination rates according to sterilization time was confirmed using seeds harvested from fields (Figure S1). First, germination rate was investigated by cultivating maize seeds in pots. Seed germination results in pots were used as a control for comparison of the germination rates of seeds sterilized by different sterilization methods. As shown in Tables 1 and S1, Hi IIA had the lowest germination rate of 31.3%, and the highest germination rate was observed in KS141 (97.9%), though seedling growth was not uniform (Figure S6). However, severe contamination of these seeds was confirmed when sterilized by a standard sterilization method (Figure S7). Therefore, the effect of seed sterilization was investigated after immersion in a 100% commercial bleach solution for 0 min, 15 min, and 30 min (Figure S2), respectively. In order to increase the seed germination rate and clearly identify where microbial growth begins, the black layers of seeds were placed at 30~45 angles by touching the medium, and germination and contamination rate were subsequently investigated (Figure S3). As shown in Figure 1, when only using distilled water for seed sterilization, all seeds were contaminated regardless of the sterilization methods. In addition, under the condition of Treatment 1 with 100% commercial bleach, all seeds were contaminated by microorganisms (Figure 1 and Tables 2 and S2). Therefore, the seeds treated with Treatment 1 were used as a control for comparison of the contamination rates of seeds sterilized by different sterilization methods. Although there are differences by variety, the immersion time of the sterilization solution decreased from 24.6% to 41.9% for 15 min treatment (Treatment 2; Tables 2 and S3) and from 17.6% to 21.1% for 30 min treatment (Treatment 3; Tables 2 and S4). The longer the immersion time, the greater the effect of seed sterilization and the more significant the reduction in microorganisms detected (Table 2). In addition, contamination of microorganisms around the seeds could be visually observed within 4 days after being plated on media, necessitating that contaminated seeds be removed quickly. These results suggest that by allowing the black layer of maize seeds to reach the medium, the nutrients and water of the medium are supplied

sufficiently in a short time, increasing seed germination (Figure S3). Also, it would have been easy to visually observe contamination by releasing contaminants together when the roots ruptured (Figure S4).

**Table 1.** Germination rate with different sterilization treatments.

| | Treatments | | | | |
|---|---|---|---|---|---|
| | Pots | Treatment 1 | Treatment 2 | Treatment 3 | Treatment 4 |
| Varieties | Germination (%) | Germination (%) | Germination (%) | Germination (%) | Germination (%) |
| Hi IIA | 31.40 ± 0.66 | 97.30 ± 0.67 *** | 97.33 ± 0.67 *** | 98.33 ± 0.33 *** | 99.22 ± 0.69 *** |
| Hi IIA × Hi IIB | 42.37 ± 5.35 | 98.78 ± 0.69 *** | 98.78 ± 0.69 *** | 98.67 ± 0.67 *** | 99.11 ± 0.69 *** |
| A188 | 63.87 ± 2.05 | 98.44 ± 0.84 *** | 98.44 ± 0.84 *** | 99.44 ± 0.51 *** | 98.56 ± 0.69 *** |
| H99 | 62.40 ± 1.00 | 98.56 ± 0.51 *** | 98.56 ± 0.51 *** | 99.11 ± 0.51 *** | 99.44 ± 0.69 *** |
| B104 | 87.93 ± 1.46 | 98.89 ± 0.51 *** | 98.89 ± 0.51 *** | 98.89 ± 1.07 *** | 98.44 ±1.84 *** |
| B73 | 93.93 ± 1.61 | 99.67 ± 0.33 *** | 99.67 ± 0.33 *** | 99.56 ± 0.51 *** | 99.22 ± 0.84 *** |
| B98 | 96.27 ± 0.85 | 99.67 ± 0.33 *** | 99.67 ± 0.33 *** | 99.11 ± 1.02 *** | 99.56 ± 0.51 *** |
| HW3 | 70.90 ± 1.11 | 98.33 ± 0.33 *** | 98.33 ± 0.33 *** | 99.00 ± 0.67 *** | 99.44 ± 0.38 *** |
| KS140 | 93.37 ± 1.40 | 98.67 ± 1.45 *** | 98.67 ± 1.45 *** | 98.89 ± 1.64 *** | 99.22 ± 1.07 *** |
| KS141 | 97.90 ± 1.35 | 99.78 ± 0.19 *** | 98.33 ± 0.33 *** | 99.56 ± 0.51 *** | 99.22 ± 1.07 *** |
| Hi IIA(♂) × B73(♀) | 40.89 ± 1.35 | 98.22 ± 1.92 *** | 99.11 ± 1.02 *** | 98.00 ± 1.33 *** | 97.78 ± 1.68 *** |
| B73(♂) × Hi IIA(♀) | 33.00 ± 1.76 | 97.78 ± 1.68 *** | 99.11 ± 0.77 *** | 98.22 ± 1.39 *** | 97.33 ± 2.31 *** |

The range of values obtained by mean ± standard deviation with 95% confidence limits (***; $p$-value < 0.001, Student's $t$-test).

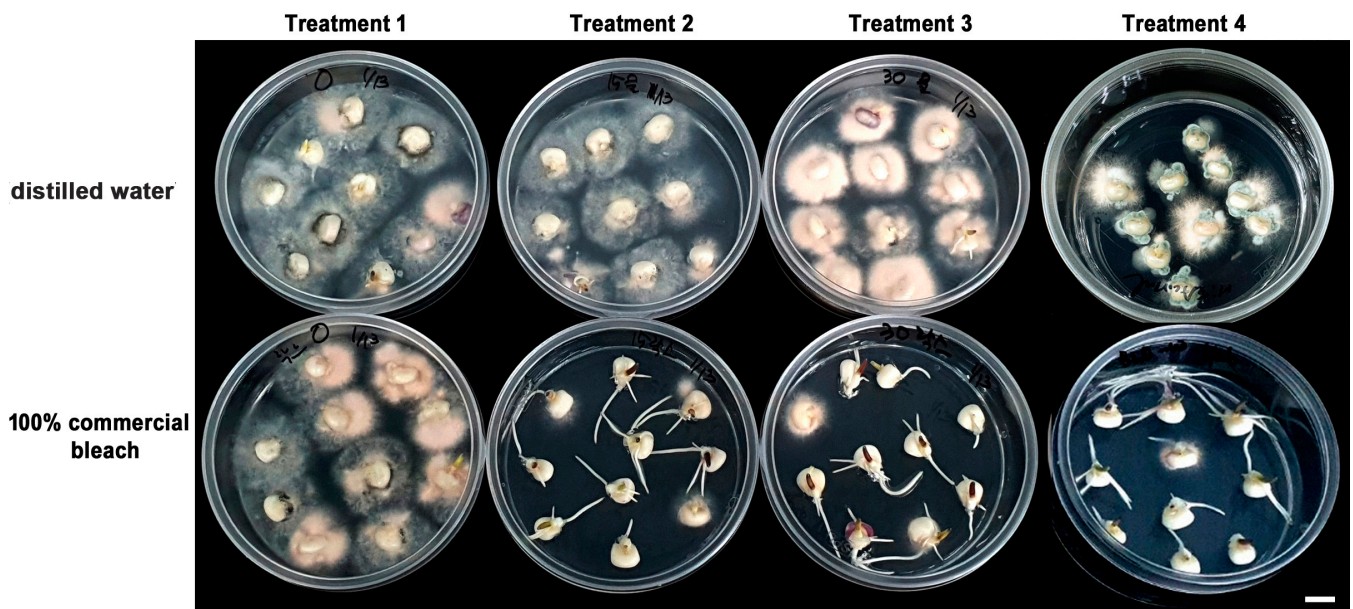

**Figure 1.** Microbial growth on maize Hi IIA seeds exposed to different sterilization treatments. Images show microbial growth on seeds exposed to sterilization treatments and grown on the basal medium for 4 days. Bar = 1 cm.

A previous study showed that increasing the sterilization time of maize seeds under shaking decreased not only the germination rate, but also the number of infected seeds [31]. However, it was reported that shaking during treatment increased the mixing of the sterilization solution and maize seeds, easily removing the air bubbles of the seeds and helping the solution penetrate sufficiently to increase seed sterilization [31]. Therefore, we investigated seed germination and contamination by adding an invert–stand–invert procedure instead of shaking in our seed sterilization method to change the sterilization procedure (Figure S2). The contamination of most seeds was 10.67% to 13.8% (Treatment 4; Tables 2 and S5), which was lower than the results of the 15 min and 30 min steril-

ization time treatments. In addition, the germination of sterilized seeds was more than 97%, indicating it does not affect germination and that the roots grow normally after germination. As a result of the investigation of seed germination and root growth after sterilization treatment, immersion time did not affect them but rather increased germination (Figure 1 and Tables 2 and S3–S5). In addition, after germination of the sterilized seeds in the medium, the growth of roots and leaves was not inhibited and they grew normally. Seedlings grown for days in the medium after seed sterilization showed almost uniform growth when transplanted into pots (Figure S8). However, our method did not completely remove microorganisms from these seeds, though it significantly reduced the detection of microorganisms and improved germination. These results suggest that the procedure involving two inversion phases and a standing phase increases the sterilization effect by increasing mixing of the sterilization solution while removing air bubbles from the surface to sufficiently wet the sterilization solution on the seed surface.

**Table 2.** Contamination rate with different sterilization treatments.

| | Treatments | | | |
|---|---|---|---|---|
| | Treatment 1 | Treatment 2 | Treatment 3 | Treatment 4 |
| **Varieties** | Contamination (%) | Contamination (%) | Contamination (%) | Contamination (%) |
| Hi IIA | 100 | 41.89 ± 1.07 *** | 19.44 ± 1.02 *** | 11.00 ± 1.67 *** |
| Hi IIA × Hi IIB | 100 | 33.78 ± 1.50 *** | 19.44 ± 1.26 *** | 11.44 ± 1.50 *** |
| A188 | 100 | 29.11 ± 4.19 *** | 17.56 ± 1.02 *** | 11.67 ± 0.88 *** |
| H99 | 100 | 26.78 ± 0.84 *** | 18.89 ± 0.19 *** | 11.33 ± 1.20 *** |
| B104 | 100 | 25.00 ± 1.20 *** | 18.56 ± 3.10 *** | 11.44 ± 2.14 *** |
| B73 | 100 | 25.67 ± 0.33 *** | 19.33 ± 2.33 *** | 10.67 ± 0.88 *** |
| B98 | 100 | 27.00 ± 0.33 *** | 20.56 ± 2.67 *** | 12.00 ± 2.03 *** |
| HW3 | 100 | 24.56 ± 1.17 *** | 19.00 ± 2.08 *** | 11.11 ± 2.22 *** |
| KS140 | 100 | 29.33 ± 1.33 *** | 19.11 ± 0.84 *** | 11.67 ± 1.53 *** |
| KS141 | 100 | 31.44 ± 3.02 *** | 21.11 ± 0.84 *** | 10.78 ± 1.68 *** |
| Hi IIA(♂) × B73(♀) | 100 | 35.78 ± 1.39 *** | 19.78 ± 2.34 *** | 13.78 ± 3.29 *** |
| B73(♂) × Hi IIA(♀) | 100 | 36.11 ± 0.84 *** | 19.33 ± 1.33 *** | 13.33 ± 0.67 *** |

The range of values obtained by mean ± standard deviation with 95% confidence limits (***; $p$-value < 0.001, Student's $t$-test).

### 3.2. Seed Sterilization Effect by Removing the Black Layer

In order to verify how the black layer affects seed contamination in seeds harvested from field-grown maize, some of the black layers of Hi IIA seeds with the highest contamination rate were removed. After sterilizing the seeds from which the black layer was removed by the method described above (Figure S5), the germination rate and contamination rate were analyzed. Seeds from which the black layer was removed showed a contamination rate of 34% when sterilized for 15 min (Treatment 2; Tables 3 and S6) and 9.4% when sterilized for 30 min (Treatment 3; Tables 3 and S6), showing 7.8% and 10% lower contamination than seeds without black layer removal (Treatment 2–3; Tables 2 and S3 and S4), respectively. The seeds containing a black layer sterilized by our newly established sterilization method showed a contamination rate of 11% (Treatment 4; Tables 2 and S5), but seeds from which the black layer was removed showed the lowest contamination rate of 0.44% (Treatment 4; Tables 3 and S6). The reason why maize seed contamination was not completely removed by our method is that other parts, including the endosperm inside the seed, were severely infected by the contaminant, and it therefore was not removed because the sterilization solution did not penetrate. However, it is considered to be a sufficiently usable method because it can identify and remove contaminated seeds that occur at a very low frequency from 3 to 4 days after germination in the medium (Figure 2).

**Table 3.** Contamination rate of Hi IIA seeds with black layer removed after exposure to different sterilization treatments.

| Variety | Treatments | | | |
|---|---|---|---|---|
| | Treatment 1 | Treatment 2 | Treatment 3 | Treatment 4 |
| | Contamination (%) | Contamination (%) | Contamination (%) | Contamination (%) |
| Hi IIA | 100 | $34.00 \pm 1.20$ *** | $9.44 \pm 1.71$ *** | $0.44 \pm 0.51$ *** |

The range of values obtained by mean $\pm$ standard deviation with 95% confidence limits (***; *p*-value < 0.001, Student's *t*-test).

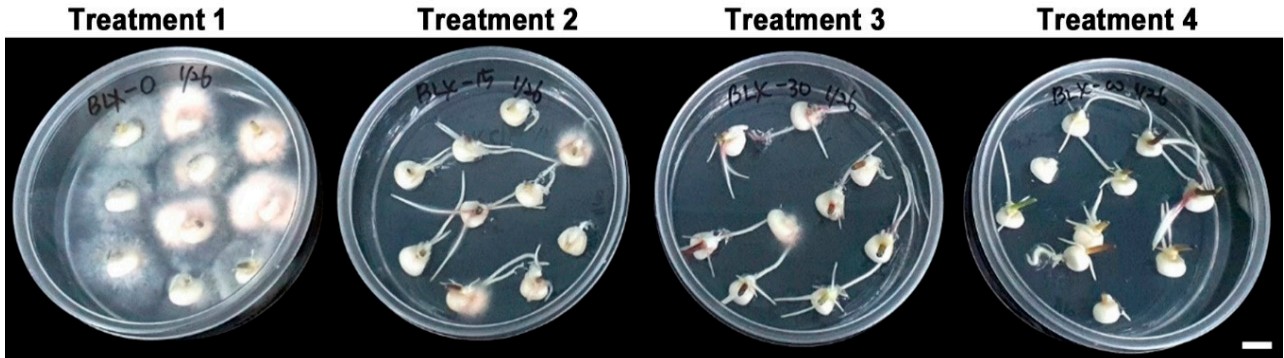

**Figure 2.** Microbial growth on maize Hi IIA seeds exposed to different sterilization treatments after removal of the black layer of maize seeds. Macroscopic images show microbial growth on seeds exposed to sterilization treatments and grown on the basal medium for 4 days. Bar = 1 cm.

At the bottom of the maize seeds, there is a milk gland at the back of the end cap, and when the kernel is fully mature, it turns black as the supply of nutrients ends [34]. The black layer generated at this time is used to determine the degree of maturity of maize because it indicates physiological maturity when harvesting for seeds [35,36]. The black layer of maize seeds is a degenerated part developed from the endosperm and contains proteins and fibers [34–36]. However, it seems that seeds were easily infected by microorganisms due to complex causes such as abnormal environments including heat waves (high temperature), drought, delayed harvested time, and storage methods. Therefore, it is believed that seed contamination through the black layer of seeds was intensified. The extremely low seed contamination rate after sterilization of seeds when the black layer is removed is a very important part in removing contamination. In addition, it is clear that removal of the black layer helped the sterilization solution to easily penetrate into the seeds through the mixing of the seeds and the sterilization solution.

### 3.3. Seed Sterilization Effect of Inverting RPM and the Number of Seeds

The high efficiency of the seed sterilization effect was confirmed when seeds were inverted at 45 RPM using the seed sterilization method outlined for Treatment 4. Therefore, it is considered that inverting RPM plays an important role in the inversion stage during the seed sterilization method of Treatment 4. In order to investigate the sterilization effect with changes to the inverting RPM, we fixed the number of Hi IIA seeds to 20 grains and measured the seed contamination rate at inverting RPM conditions of 25, 45, and 65 RPM, respectively. As a result of the analysis, the seed contamination rates were 16.56%, 0.22%, and 12.67% under the inverting RPM conditions of 25, 45, and 65 RPM (Figure 3A and Tables 4 and S7), respectively. From these results, it is considered that air bubbles in the seeds were effectively removed through mixing of the sterilization solution and the seeds under the inverting RPM condition of 45 RPM, and the highest seed sterilization effect was shown when sufficiently soaking the sterilization solution on the surface of the seeds [31]. The results indicate that the optimum sterilization effect with the lowest contamination

rate was shown when maize seeds were sterilized under the inverting RPM condition of 45 RPM.

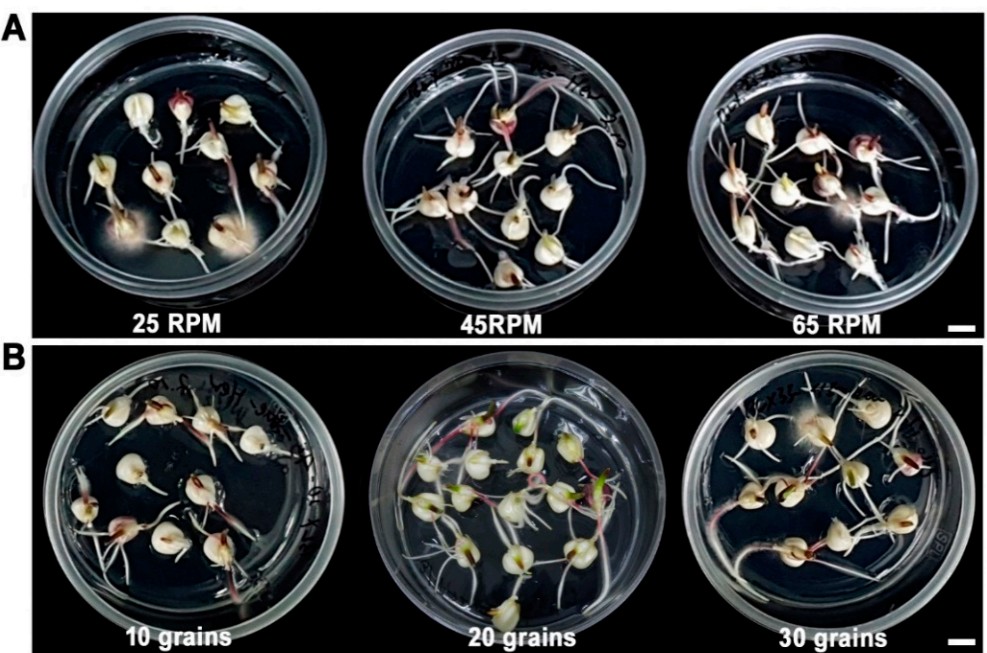

**Figure 3.** Microbial growth with changing inversion RPM and different numbers of seeds during seed sterilization according to the method of Treatment 4 after removal of the black layer of maize in Hi IIA seeds. Photographs show the growth of microorganisms on seeds subjected to different RPM (**A**) and with differing numbers of seeds (**B**) during sterilization treatment (seeds were grown on basal medium for 4 days). Bars = 1 cm.

**Table 4.** Effects of changes to inversion RPM and seed number during sterilization treatment 4 of Hi IIA seeds with black layer removed.

| Parameters | | Treatment 1 | Treatment 4 + Black Layer Removed |
|---|---|---|---|
| | Conditions | Contamination (%) | Contamination (%) |
| Inverting RPM | 25 | 100 | 16.56 ± 0.69 *** |
| | 45 | 100 | 0.22 ± 0.38 *** |
| | 65 | 100 | 12.67 ± 0.88 *** |
| Number of seeds (grains) | 10 | 100 | 0.22 ± 0.38 *** |
| | 20 | 100 | 0.22 ± 0.19 *** |
| | 30 | 100 | 10.22 ± 0.51 *** |

The range of values obtained by mean ± standard deviation with 95% confidence limits (***; $p$-value < 0.001, Student's $t$-test).

In the seed sterilization treatment, the ratio of the sterilization solution to the number of seeds was also considered an important factor in determining the sterilization effect. In order to investigate the sterilization effect in relation to the number of seeds, we put 10, 20, and 30 grains in 35 mL of sterilization solution and sterilized them using the seed sterilization method outlined for Treatment 4 to measure the seed contamination rate. When 10 grains and 20 grains were used, the contamination rate was 0.22%, and when 30 grains were used, the contamination rate increased to 10.22% (Figure 3B and Tables 4 and S8). There was no significant difference in contamination rate results between the conditions of using 10 grains and 20 grains. It can be seen that the sterilization solution used when sterilizing 10 grains was used excessively, and it is not an optimal condition. In the case of 30 grains, the increase in contamination rate is considered to be due to the fact that the

sterilization solution and seeds were not sufficiently mixed during sterilization treatment, and the sterilization solution did not wet the surface of the seeds sufficiently due to the remaining air bubbles. These results suggest that the ratio of sterilization solution to the number of seeds is an important factor affecting seed sterilization. Therefore, it was confirmed that in our newly improved maize seed sterilization method, when 20 grains are used in 35 mL of sterilization solution, the optimal conditions for the highest sterilization effect are observed.

From the sterilization results of Hi IIA seeds from which the black layer was removed, it was verified that 11 varieties also showed similar sterilization effects. As shown in Tables 4 and S9, all the seeds from which the black layer was removed showed a low contamination rate of 0% to 0.44% (average 0.29%). Interestingly, microbial contamination could not be observed until the 10th day after germination in the medium (Figure S9), and even when the germinated seeds were transplanted into the field, this did not affect the growth of roots and leaves (Figure S10). Therefore, it is considered that the sterilization effect is increased without growth inhibition due to the removal of the black layer. These results suggest that pollutants can be almost completely removed and that germination can be induced with high efficiency by removing the black layer of maize seeds. Although it is not clear whether it directly or indirectly affected our results, the transformation efficiency of Hi IIA immature embryos cultivated from seeds with the black layer removed and sterilized according to the method of Treatment 4 was 3.3 times higher than that of maize grown from seeds disinfected using the method of Treatment 2 (Tables 5 and S10). These results suggest that transformation efficiency can also be improved by a new seed sterilization method.

**Table 5.** Contamination rate of 11 varieties of seeds with the black layer removed after exposure to sterilization using the method of Treatment 4.

| Varieties | Treatments | |
|---|---|---|
| | Treatment 1 | Treatment 4 + Black Layer Removed |
| | Contamination (%) | Contamination (%) |
| Hi IIA × Hi IIB | 100 | 0.44 ± 0.19 *** |
| A188 | 100 | 0.44 ± 0.38 *** |
| H99 | 100 | 0.44 ± 0.19 *** |
| B104 | 100 | 0 *** |
| B73 | 100 | 0 *** |
| B98 | 100 | 0 *** |
| HW3 | 100 | 0 *** |
| KS140 | 100 | 0.44 ± 0.19 *** |
| KS141 | 100 | 0.33 ± 0.33 *** |
| Hi IIA(♂) × B73(♀) | 100 | 0.44 ± 0.38 *** |
| B73(♂) × Hi IIA(♀) | 100 | 0.44 ± 0.38 *** |

The range of values obtained by mean ± standard deviation with 95% confidence limits (***; $p$-value < 0.001, Student's $t$-test).

Finally, our results show that the newly improved maize seed sterilization method can easily and inexpensively increase seed sterilization and germination rates for maize seeds harvested from the field through the use of a commercially available disinfectant.

## 4. Conclusions

This study established and reperformed an appropriate seed sterilization method using disinfectant for field-grown and harvested maize seed. Our study used 100% commercial bleach disinfectant (4–5% NaClO) for the sterilization of maize seeds (20 grains) from which the black layer had been removed, and the addition of the invert (45 RPM)–stand (0 RPM)–invert (45 RPM) procedure resulted in significantly reduced microbial contamination. In addition, it was possible to obtain seedlings that grew without microorganisms through

sterilization. Our sterilization method showed that the remaining microbial substances cannot grow any more after 7 days due to the removal of microorganisms from the seeds. Also, surface sterilization treatment does not seem to affect the growth of roots and leaves. Overall, the analyses performed provide evidence that our seed sterilization method is the most effective treatment for field-harvested maize seed and can be used for further studies such as genome editing and maize transformation.

**Supplementary Materials:** The following supporting information can be downloaded at: https://www.mdpi.com/article/10.3390/agriculture13112147/s1, Figure S1: Phenotypes of maize Hi IIA seeds harvested from fields in July and August, Figure S2: The applied methods of maize seed sterilization, Figure S3: Placement and growth of sterilized maize seeds on the MS medium, Figure S4: Germination of maize Hi IIA seeds after exposure to sterilization treatment, Figure S5: Location and removal area of the black layer in maize Hi IIA seeds, Figure S6: Standard flat 50-hole pots filled with potting mix for the planting of maize Hi IIA seeds, Figure S7: Microbial growth on maize Hi IIA seeds sterilized by the standard method, Figure S8: Maize Hi IIA seedlings grown for 4 days in the medium after exposure to sterilization according to the method of Treatment 4 which were transplanted into pots and grown in growth chamber for 7 days, Figure S9: Maize Hi IIA seedling growth after exposure to sterilization according to the method of Treatment 4, Figure S10: Greenhouse growth after disinfecting maize seeds with the optimized sterilization method, Table S1: Germination of maize seeds in pots, Table S2: Contamination and germination of maize Hi IIA seeds after exposure to sterilization according to the method of Treatment 1, Table S3: Contamination and germination of maize Hi IIA seeds after exposure to sterilization according to the method of Treatment 2, Table S4: Contamination and germination of maize Hi IIA seeds after exposure to sterilization according to the method of Treatment 3, Table S5: Contamination and germination of maize Hi IIA seeds after exposure to sterilization according to the method of Treatment 4, Table S6: Contamination and germination of maize Hi IIA seeds during different sterilization treatments after having the black layer removed, Table S7: Effects of changes to inversion RPM during maize seed sterilization according to the method of Treatment 4 with removal of the black layer of maize in Hi IIA seeds, Table S8: Effect of the number of seeds during maize seed sterilization according to the method of Treatment 4 with removal of the black layer of maize in Hi IIA seeds, Table S9: Contamination and germination of seeds after exposure to sterilization according to the method of Treatment 4 with removal of the black layer of maize seeds, Table S10: Transformation efficiency according to sterilization method in Hi IIA maize seeds.

**Author Contributions:** J.K.H. and E.J.S. conceived and designed the study. J.K.H., J.B., S.R.P., G.S.L. and E.J.S. performed all experiments. J.K.H., G.S.L. and S.R.P. collected and analyzed the data. J.K.H. and J.B. were involved in strategic planning. J.K.H. and J.B. wrote the manuscript. All authors have read and agreed to the published version of the manuscript.

**Funding:** This work was carried out with support from the Research Program for Agricultural Biotechnology (RS-2022-RD009520), National Institute of Agricultural Sciences, Rural Development Administration, Republic of Korea. This study was supported in 2023 by the RDA Fellowship Program of the National Institute of Agricultural Sciences, Rural Development Administration, Republic of Korea.

**Institutional Review Board Statement:** Not applicable.

**Data Availability Statement:** No new data were created or analyzed in this study. Data sharing is not applicable to this article.

**Conflicts of Interest:** The authors declare no conflict of interest.

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
