# Peer review of "A New Protocol to Mitigate Damage to Germination Caused by Black Layers in Maize (Zea mays L.) Seeds"

_agriculture, doi:10.3390/agriculture13112147_

Round 1

Reviewer 1 Report

Comments and Suggestions for Authors

The manuscript “Establishment of efficient sterilization method by removing the black layer from maize (Zea mays L.) seeds harvested in field” is an interesting study. Just a few recommendations to the authors:

 1-      In the Abstract, the abbreviation RPM is not explained. 

2-      In the Introduction section, the authors wrote several times “sterile seeds” and “seed sterilization”; and although it is evident that they refer to the surface of the seed, it is recommended that they specify it in each instance (also this is present in other sections). The authors wrote “ethanol is higher than that of ethanol only”, it is recommended that the authors explain the phrase better. Also, the authors wrote “due to many microorganisms present in the population”, but what populations of microorganisms are the authors referring to? 

3-      In Materials and methods section, it is recommended that the authors specify which institution provided each variety, since it is not specified in this section or in the supplementary material. Also, it is recommended to the authors explain better the procedure of the phrase “Maize seeds were immersed enough to be submerged and germinated at 25ºC. Three”. The authors wrote: “it was transferred to a 50-hole pot”, but are they refer to the seeds? Figure S2 could be improved and more explanatory with images or drawings. In figures S3-S10 the plant material used is not mentioned, and there is no bar associated with a measurement scale. The relationship of Figures S3 and S4 with the information where them are  indicated is not clear. The authors wrote: “the seeds, seeds were visually investigated 7 days after placing into the media”, Could the authors explain what variables they measured?

4-      In Results and Discussion section, the authors wrote “sterilized at the same time using sterile water were used as a control group” however, the word sterilization should not be used to establish control. Table 1, 2 and 4 should be improved to make the data clear. In Figures 1, 2, and 3, the plant material used is not mentioned. In all the figures there is no bar associated with a measurement scale. Did the authors identify at least at the genus level and with molecular techniques ,the microorganisms found in the seeds, or those microorganisms that were persistent despite the sterilization treatment of the surface of the seeds? Did they manage to locate these microorganisms somewhere in the seeds?

Author Response

Title: Establishment of efficient sterilization method by removing the black layer from maize (Zea mays L.) seeds harvested in field

Ms. Ref. No.: Agriculture-2673631

Author(s): Joon Ki Hong et al.

Dear Review;

Thank you for your review of the manuscript. We have addressed the reviewer’s concerns, questions, and suggestions, and revised the manuscript accordingly.

In addition, we have made several modifications to clarify and strengthen the manuscript as proposed by the reviewers. The following manuscript is detailed responses to the referees’ comments.

Sincerely yours,

Eun Jung Suh, Ph.D.

Gene Engineering Division, National Institute of Agricultural Sciences, Rural evelopment Administration, 370 Nongsaengmyeong-ro, Jeonju 54874, Republic of Korea

E-mail : [email protected]

Tel.: +82-63-238-4660

Reviewer 2 Report

Comments and Suggestions for Authors

Manuscript code: 2673631

Manuscript title: Establishment of efficient sterilization method by removing the black layer from maize (Zea mays L.) seeds harvested in field

The success of tissue culture and genetic transformation strongly depends on the sanitary quality of the initial plant material. In this study the authors proposed a new protocol for a proper sterilization of maize seeds. The manuscript is overall well written. Below are some corrections and comments that need to be addressed, should the paper be accepted.

Overall comments

There is no line numbering. This complicated the review.

Introduction

Lines 2-4: Replace: “Seeds are attached or inhabited by many bacteria and fungi, and some cause a lot of damage to germination, seedlings, and plant growth” by “Seeds are colonized by many bacteria and fungi, and some may be harmful for the germination and subsequent seedlings and plant growth”.

Line 5-7: Rephrase the sentence: “In particular, in order to apply the latest technology such as genome-editing, in vitro tissue culture and genetic transformation techniques require a sterile working environment and contaminant free starting materials”. Not clear as it currently stands.

Rephrase the sentence: “Shaking added during sterilization reported that the mixing of sterilization solution and maize seeds increased inhalation, helped remove air bubble from seeds, and sufficiently wetted the sterilization solution on the seed surface”

“Also, the climate change can cause contamination by various pathogens during the maize cultivation period [31].”: How? This needs clarification.

Materials and methods

Ten maize varieties were obtained from NICS (National Institute of Crop Science, Rural Development Administration, 181 Hyeoksin-ro, Iseo-myeon, Wanju-Gun, Jeollabuk-do, Korea), MES (Maize Experiment Station, Gangwondo Agricultural Research and Extension Services, Korea), and USDA (United States Department of Agriculture), and two major hybrid varieties were made and used through Hi IIA and B73 hybridization. Yes, but present in a table the list of each of the maize varieties and their origin.

The statistical analyses were poorly detailed. How was treatment effect checked? What statistical tests were employed, and to compare what? And what was the decision threshold?

Results

All the table in the manuscript needs statistic support (statistical test employed, degree of freedom, p-value, letters indicating statistical grouping).

Without the elements above mentioned, I fear I will not be able to properly assess the results.

Comments on the Quality of English Language

The English needs to be polished!

Author Response

(The authors gave the same response as above.)

Reviewer 3 Report

Comments and Suggestions for Authors

1.    The Language and grammar of the manuscript needs revision.

2.    Section 2.1: Please clearly specify the time zone while referring to the time of pollination.

3.    The label of figures and their reference in the running text do not match. Nowhere figure S1 to S7, S9, S10 and Table S9 can be located in the manuscript.

4.    Section 3.1; Paragraph 2: What do you mean by “Seeds sterilized at the same time using sterilized water”? Can sterile water be used for sterilization of seeds?

5.    Tables 1 and 2: Where is the control? Is it Treatment 1? Please mention it clearly and separately from other treatments.

Comments on the Quality of English Language

Please refer to the previous section

Author Response

(The authors gave the same response as above.)

Reviewer 4 Report

Comments and Suggestions for Authors

The research is very simple and advances very little scientific knowledge; however, the evaluated method can be used successfully to mitigate the negative effects of the black layer in maize seeds. Some points below should be reviewed before potential publication of the manuscript:

1. The title is very redundant in several aspects; my suggestion is that it can be objective and clear; this is just a suggestion: “A new protocol to mitigate damage to germination caused by darker layer in maize (Zea mays L.) seeds”

2. In the abstract, instead of starting by talking about the importance of corn... The authors can start by talking about the problem. What is the negative impact of the black layer on maize seeds? What causes this?

3. The introduction is very long and does not make clear the main focus of the study... If the objective is to have seeds with high health for plant tissue culture, would it not be interesting to report in the introduction that there are problems in the in vitro establishment of plant seeds? corn because of contamination? Is this the problem with the study? Could this disinfestation protocol be applied for purposes other than plant tissue culture? I recommend that the authors improve the introduction of the manuscript; making it clearer and more objective.

4. In the methodology, authors are encouraged to insert data on climatic conditions during field cultivation (e.g., temperature average, maximum, and minimum; relative humidity; vapor pressure deficit; and photosynthetically active radiation).

5. The description of treatments needs to be better described in the methodology. This section was really bad. Describe in order the four treatments and what each one represents in terms of protocol; This is important so that the reader can understand the procedures.

6. Table 1 is unformatted; it was not possible to understand. Please check the magazine's rules. Additionally, you can remove the term “rate” and keep only Germination (%) or Contamination (%).

7. Germination and contamination values can be separated into separate tables.

8. Table 2 can be transformed into a graph to better visualize the response pattern.

Overall, the work reports that the protocol can be used in plant tissue culture; however, the authors did not test the application of the protocol in tissue culture. Throughout the text, this issue of tissue culture can even be removed to avoid leaving the reader confused about the objective of the study. Initially, the text led to an understanding; but in the end, it became clear that this was not tested.

Author Response

(The authors gave the same response as above.)

Round 2

Reviewer 4 Report

Comments and Suggestions for Authors

The authors did a good job while reviewing the manuscript. I recommend that it be accepted for publication in the agriculture journal.